# Expression of Iron Metabolism Proteins in Patients with Chronic Heart Failure

**DOI:** 10.3390/jcm11030837

**Published:** 2022-02-05

**Authors:** Bogna Kozłowska, Barbara Sochanowicz, Leszek Kraj, Małgorzata Palusińska, Piotr Kołsut, Łukasz Szymański, Sławomir Lewicki, Witold Śmigielski, Marcin Kruszewski, Przemysław Leszek

**Affiliations:** 1Department of Heart Failure and Transplantology, The Cardinal Stefan Wyszyński National Institute of Cardiology, Alpejska 42, 04-628 Warsaw, Poland; bkozlowska@ikard.pl; 2Centre of Radiobiology and Biological Dosimetry, Institute of Nuclear Chemistry and Technology, Dorodna 16, 03-195 Warszawa, Poland; b.sochanowicz@ichtj.waw.pl (B.S.); marcin.kruszewski@ichtj.waw.pl (M.K.); 3Department of Oncology, Medical University of Warsaw, 01-163 Warsaw, Poland; l.kraj@igbzpan.pl; 4Department of Molecular Biology, Institute of Genetics and Animal Biotechnology, Polish Academy of Science, Postępu 36A, 05-552 Magdalenka, Poland; m.palusinska@igbzpan.pl (M.P.); l.szymanski@igbzpan.pl (Ł.S.); s.lewicki@igbzpan.pl (S.L.); 5Department of Cardiac Surgery and Transplantology, The Cardinal Stefan Wyszyński National Institute of Cardiology, Alpejska 42, 04-628 Warsaw, Poland; pkolsut@ikard.pl; 6Faculty of Medical Sciences and Health Sciences, Kazimierz Pulaski University of Technology and Humanities, 26-600 Radom, Poland; 7Department of Epidemiology, Cardiovascular Disease Prevention and Health Promotion, The Cardinal Stefan Wyszyński National Institute of Cardiology, Alpejska 42, 04-628 Warsaw, Poland; wsmigielski@ikard.pl; 8Department of Molecular Biology and Translational Research, Institute of Rural Health, Jaczewskiego 2, 20-090 Lublin, Poland

**Keywords:** heart failure, myocardial iron metabolism, oxidative stress, myocardial gathering proteins expression, human model

## Abstract

In heart failure, iron deficiency is a common comorbid disease that negatively influences exercise tolerance, number of hospitalizations and mortality rate, and this is why iron iv supplementation is recommended. Little is known about the changes in iron-related proteins in the human HF myocardium. The purpose of this study was to assess iron-related proteins in non-failing (NFH) vs. failing (FH) human myocardium. The study group consisted of 58 explanted FHs; control consisted of 31 NFHs unsuitable for transplantation. Myocardial proteins expressions: divalent metal transporter (DMT-1); L-type calcium channel (L-CH); transferrin receptors (TfR-1/TfR-2); ferritins: heavy (FT-H) or light (FT-L) chain, mitochondrial (FT-MT); ferroportin (FPN), regulatory factors and oxidative stress marker: 4-hydroxynonenal (4-HNE). In FH, the expression in almost all proteins responsible for iron transport: DMT-1, TfR-1, L-CH, except TfR-2, and storage: FT-H/-L/-MT were reduced, with no changes in FPN. Moreover, 4-HNE expression (pg/mg; NFH 10.6 ± 8.4 vs. FH 55.7 ± 33.7; *p* < 0.0001) in FH was increased. HNE-4 significantly correlated with DMT-1 (r = −0.377, *p* = 0.036), L-CH (r = −0.571, *p* = 0.001), FT-H (r = −0.379, *p* = 0.036), also FPN (r = 0.422, *p* = 0.018). Reducing iron-gathering proteins and elevated oxidative stress in failing hearts is very unfavorable for myocardiocytes. It should be taken into consideration before treatment with drugs or supplements that elevate free oxygen radicals in the heart.

## 1. Introduction

Heart failure (HF) is commonly accompanied by iron deficiency (ID), which is associated with depleted body iron stores, reduced systemic iron availability and unmet cellular iron requirements [1]. On the cellular level, iron is one of the critical microelements involved in diverse metabolic processes, including oxygen transport, synthesis of deoxyribonucleic acid and oxidative energy production [2]. Systemic ID is associated with reduced exercise tolerance, increased symptom severity and higher mortality rates independent of coexisting anemia. Randomized clinical trials have shown that in the case of ID, intravenous iron supplementation improves symptoms, exercise capacity and quality of life in ambulatory patients with chronic HF [3,4]. Very recently, Ponikowski et al. showed that in patients with ID who were stabilized after an episode of acute HF, *iv.* iron treatment reduced the risk of heart failure hospitalizations [5]. According to the 2021 ESC HF Guidelines, in such groups of patients, iron replacement therapy should be considered [6]. Following this recommendation, iron iv supplementation has become increasingly frequent in HF patients presenting ID.

However, little is known about the changes in proteins responsible for myocardial iron turnover in human HF. Therefore, proper characterization of iron homeostasis in HF seems important as additional iron supplementation could potentially exert a harmful effect related to the production of intracellular reactive oxygen species (ROS). In addition, iron-induced oxidative damage can alter myocyte function, directly affecting the function of several excitation–contraction coupling proteins, increasing myocyte loss and interstitial fibrosis [7]. Moreover, elevated iron may also affect the function of endothelial and smooth muscle cells, which can additionally affect myocardial perfusion and function [8,9].

Thus, the purpose of our study was to assess the changes in proteins related to iron in a failing human myocardium. In addition, the concentration of iron-handling proteins, their regulatory factors and an oxidative stress marker was evaluated in the non-failing and failing human myocardium.

## 2. Materials and Methods

### 2.1. Study Population and Protocol

The protocol was approved by the Local Ethics Committee (IK-NPIA-0021-34/1699/18, from 15 May 2018). Each patient participating in the study signed an informed consent form after a detailed explanation of the principles. The study group comprised 58 consecutive patients referred to heart transplantation (OHT). Myocardial studies were performed on the failing ventricular myocardium (FH) obtained during transplantation. Before transplantation, all patients with chronic kidney disease stage G5 (GFR < 15 mL/min/1.73 m^2^) and other clinically significant comorbidities such as known gastroenteral bleeding, infections and antibiotic therapy were excluded from the study. Myocardium from 31 non-failing heart (NFH) subjects (11 women and 20 men) aged 18–54 years, who had died from head trauma and were unsuitable for heart transplantation due to hepatitis B, CMV infection, extensive damage during harvest or size donor/recipient mismatch, was used as a control for comparison with FH myocardium. Donors from the control group did not have chronic diseases, which was confirmed with family members of the deceased. In this group, the leading cause of death was subarachnoid bleeding.

### 2.2. Study Protocol

Before OHT, all patients were stratified according to the New York Heart Association (NYHA) classification and made to undergo a clinical assessment that included echocardiography, right heart catheterization and a laboratory test.

*Rest Echocardiography*—Two-dimensional, M-mode and color Doppler transthoracic echocardiography were performed at rest in the left lateral position using commercially available equipment (Vingmed V) and standard views. Left ventricle (LV) diameters—diastolic (LVEDD) and systolic (LVESD)—ejection fraction (LVEF), right ventricle diastolic size (RV), right ventricular systolic pressure (R*VS*P), tricuspid annular plane systolic excursion (TAPSE) and others were measured/calculated according to the recommendations of the American Society of Echocardiography.

*Right heart catheterization*—Hemodynamic and cardiac output measurements, together with vascular resistance and transpulmonary gradient calculation, were performed just before OHT.

#### Laboratory Test

*Blood counts* were determined with an automatic counter (Sysmex K4500, Norderstedt, Germany): Red Blood Cells (RBC normal ranges—male/female—4.6–6.2/4.2–5.4 mln/µL), hemoglobin (Hb—14–18/12–16 g/dL), hematocrit (Hct—42–52/37–47%), mean corpuscular volume (MCV—80–99 fl), mean corpuscular hemoglobin (MCH—27–32 pg), mean corpuscular hemoglobin concentration (MCHC—32.2–35.5 g/dL), red blood cell distribution width standard deviation (RDW-SD—35.1–43.9 fL), red blood cell volume distribution width (RDW-CW—11.6–14.4%), white blood cells (WBC 4.2–9.1 10^3^/µL), platelet (PLT 150–400 10^3^/µL), mean platelet volume (MPV—9.8–16.1 fL).

*Reticulocyte parameters*: reticulocyte (RET%—0.48–1.64%), reticulocyte count (RET—0.026–0.078 mln/µL) immature reticulocyte fraction (IRF—86.5–98.5%), Low Fluorescence Ratio (LRF—86.5–98.5), Medium Fluorescence Ratio (MFR—1.5–11.3%), High Fluorescence Ratio (HFR- 0.0–1.4%), Reticulocyte hemoglobin equivalent (RET-Hb—28–33 pg) were measured with Sysmex XN 550.

*Body iron status and biochemical assessment* were evaluated in serum by measuring with Clinical Chemistry System Olympus 680 (Beckman Coulter, Brea, CA, USA)—serum iron (normal ranges—male/female 70–180/60–180 µg/dL), TIBC (calculated from iron—210–340/260/390 µg/dL), transferrin (200–360 mg/dL), transferrin saturation (TSAT calculated from serum iron/transferrin—15–45%); with ARCHITECT^®^ Immunochemistry Diagnostics Platform (Abbott Laboratories, Chicago, IL, USA)—ferritin (FR—normal range male/female 4.63–204/21.81–274.66 ng/mL), folic acid (2.34–17.56 ng/mL), vitamin B12 (189–883 pg/mL); with COBAS C501 (Roche Diagnostic, Basel, Switzerland)—C-reactive protein (CRP—normal ranges—0–0.5 mg/dL); sodium (136–145 mmol/L), potassium (3.5–5.1 mmol/L) bilirubin (≤17.1 µmol/L), total proteins (66–87 g/L), uric acid (3.4–7.0 mg/dL), aspartate transaminase (Ast—normal range male/female 0–40/0–32 U/L), alanine transaminase (Alt—0–41/0–33 U/L), creatinine (62–106/44–80 µmol/L), soluble transferrin receptor (sTfR—2.2–5.0/1.9–4.4 mg/L); glucose (70–99 mg/dL), with Chemiluminescent IMMULITE 2000 (Siemens Healthcare Diagnostics, Marburg, Germany)—erythropoietin (EPO—3.7–29.5 mIU/mL) or with Cobas e411R analyzer (Roche Diagnostic, Basel, Switzerland)—N-terminal pro B-type natriuretic peptide (NT-proBNP—normal ranges 0–125 pg/mL). Creatinine clearance was estimated by the Cockcroft–Gault formula.

### 2.3. Preparation of Cardiac Tissue

Tissue samples of the left ventricular free walls were taken at the time of explantation (avoiding scarred, fibrotic or adipose tissue, endocardium, epicardium or great vessels), rinsed immediately, blotted dry, frozen in liquid nitrogen, and kept at −80 °C until use.

### 2.4. Assessment Iron-Related Proteins

Cardiac samples of 50–100 mg were homogenized using an Omni TH International (Omni International, Kennesaw, GA, USA) homogenizer in a buffer (10 μL/1 mg of tissue) containing: 20 mM HEPES pH 7.9 with 1.5 mM MgCl_2_, 10 mM KCl, 0.5 mM EDTA, 2% glycerol, 0.5% sodium deoxycholate, 0.5% NP-40 (human), Complete Protease Inhibitor Cocktail and 40 mM sucrose. Homogenate was centrifuged at 20,000× *g* for 20 min, the supernatant was collected, portioned, frozen in liquid nitrogen, then stored in the freezer at −85 °C. The total protein concentrations were determined with the Bradford method.

Myocardial iron-gathering proteins concentrations were assayed by ELISA according to the manufacturer’s instructions (ELK Biotechnology Co., Ltd., Wuhan, China): divalent metal transporter—(DMT-1), L-type calcium channel (L-CH), transferrin receptor -1/2 (TfR-1/TfR-2), ferritin heavy chain (FT-H), ferritin light chain(FT-L), ferritin mitochondrial (FT-MT), ferroportin (FPN), hepcidin (Hepc), aconitase 1 (ACO-1), iron response protein 2 (IREB-2), hypoxia-induced factor (HIF-1), hemojuvelin (HJV), 4-hydroxynonenal (4-HNE). When necessary, cardiac samples were diluted according to the manufacturer’s recommendation (Appendix A).

### 2.5. Statistical Analysis

An analysis of the distribution of continuous variables using the W Shapiro–Wilk test showed that the studied variables (mostly) did not follow normal distribution, and so the Mann–Whitney test U was applied by comparing two groups and the Kruskal–Wallis ANOVA test was applied by comparing three or more groups (with relevant post-hoc test). Analysis of quality variables was performed by chi-square test. Correlation between quantitative variables characterized by normal distribution was calculated either by using Pearson coefficient or Spearman coefficient. All statistical calculations were performed using the STATISTICA 12 software (StatSoft Polska Sp. z o.o.; Kraków, Poland). The significance level was set as *p* < 0.05.

## 3. Results

### 3.1. Clinical Characteristic of the Study Groups (Heart Failure and Non-Failing Subjects)

The study group consisted of 58 consecutive patients (13 women, 45 men), mean age 53.5 years, who underwent heart transplantation (Failing Heart group—FH) due to advanced HF. The primary HF etiology was idiopathic cardiomyopathy (*n* = 36) and coronary artery disease (*n* = 18). Forty patients presented sinus rhythm, while the rest (*n* = 18) had atrial fibrillation/flutter. Despite optimal pharmacological therapy consisting of angiotensin-converting enzyme inhibitors/ARNI, aldosterone antagonists, β-blockers and diuretics, the patients remained symptomatic, classified accordingly as NYHA functional class III (*n* = 31) and class IV (*n* = 27).

The study group presented with LV dilatation (LVDD 67 ± 8 mm; L*VS*D 59 ± 13 mm), together with reduced contraction (LVEF 20 ± 7.5%), RV enlargement (RVD 42 ± 6.5 mm) and pulmonary hypertension (pulmonary vascular resistance 2.21 ± 0.93 W.u). The patients also demonstrated significant neurohumoral (NTproBNP 3955.5 ± 2250.5 pg/mL) activation.

In the hematological and biochemical assessment, the study group did not reveal any significant abnormalities. Furthermore, no significant differences were found between men and women in the assessed parameters within the study group, except the parameters related to height and body weight (Table 1, S2 for FH group, Appendix A for NHF group S3).

### 3.2. Expression of Iron-Handling Proteins, Their Regulatory Factors and an Oxidative Stress Marker in the Non-Failing Compared to the Failing Human Myocardium

Compared to NFH, in FH we observed a significant reduction in almost all proteins responsible for iron transportation into the cell: DMT-1 (ng/mg; NFH 5.8 ± 1.1 vs. FH 4.3 ± 0.6; *p* < 0.0001); L-CH (ng/mg; NFH 67.3 ± 23.2 vs. FH 47.3 ± 8. 5; *p* = 0.0019); TfR-1 (ng/mg; NFH 173.0 ± 36.5 vs. FH 152.0 ± 30.8; *p* = 0.0038), except TfR-2. In addition, in FH a significant reduction in expression of proteins responsible for iron storage was observed: FT-H (ng/mg; NFH 28.1 ± 5.0 vs. FH 23.5 ± 3.8; *p* = 0.0189); FT-L (ng/mg; NFH 22.1 ± 7.4 vs. FH 15.0 ± 5.7; *p* < 0.0001); and FT-MT (ng/mg; 55.9 ± 12.4 vs. 39.1 ± 6.1; *p* < 0.0001), as compared to NFH. Interestingly, there was no significant change in the expression of the protein responsible for iron removal from the cell—FPN.

Among regulatory factors, we found a significant reduction in myocardial expression of several regulatory factors: ACO-1 (ng/mg; NFH 58.1 ± 8.6 vs. FH 50.0 ± 5.4; *p* = 0.0045), IREB-2 (ng/mg; NFH 1.0 ± 0.3 vs. FH 0.7 ± 0.1; *p* = 0.0104), HIF-1 (ng/mg; NFH 11.8 ± 4.2 vs. FH 7.9 ± 2.2; *p* = 0.0007), except Hepc and HJV.

Finally, there was a significant increase in oxidative stress marker 4-HNE in FH compared to NFH (pg/mg; NFH 10.6 ± 8.4 vs. FH 55.7 ±33.7; *p* < 0.0001) (Figure 1, Appendix A).

#### Expression of Iron-Handling Proteins in the Non-Failing Compared to the Failing Human Myocardium (Non-Iron Deficient and Iron Deficient)

In a further analysis, we stratified our population into two subgroups utilizing the most frequently used parameters to characterize iron deficiency—ferritin and TSAT: ferritin < 100 or ferritin 100–299 ng/mL and TSAT < 20—iron deficient (HF- ID) and ferritin > 300 ng/mL—non iron deficient (HF-NID) subjects. Within the HF group, upon comparing FH-ID vs. FH-NID subjects, we did not found any significant changes in the expression of proteins responsible for iron transportation (Appendix A). As in the entire FH group, in both subgroups also we confirmed a significant reduction in almost all proteins responsible for iron transportation into the cell: DMT-1 (ng/mg; NFH 5.8 ± 1.1 vs. FH-NID 4.2 ± 0.5 vs. FH-ID 4.4 ± 0.6; *p* < 0.0001); L-CH (ng/mg; NFH 67.3 ± 23.2 vs. FH-NID 43.1 ± 7.4; *p* < 0.004); TfR-1 (ng/mg; NFH 173.0 ± 36.5 vs. FH-NID 153.5 ± 31.22; *p* = 0.0093); also proteins responsible for iron storage FT-L (ng/mg; NFH 22.1 ± 7.4 vs. FH-NID 17.0 ± 6.5 vs. FH-ID 12.2 ± 5.5; *p* < 0.0002); and FT-MT (ng/mg; NFH 55.9 ± 12.4 vs. FH-NID 34.8 ± 6.0 vs. FH-ID 44.0 ± 4.4; *p* < 0.0001). There was no significant change in expression of TfR-2, either responsible for iron removal from the cell–FPN (Appendix A).

### 3.3. Correlations between Iron-Handling Proteins, Regulatory Factors and Oxidative Stress Markers in the Non-Failing Human Myocardium

In NFH, we found several correlations among assessed proteins, regulatory factors and oxidative stress.

#### 3.3.1. Relations among Iron Regulatory Proteins

Among the proteins responsible for iron transportation into the cell, a positive correlation between TRF-1 and L-CH (r = 0.5289, *p* = 0.005) was observed. Moreover, L-CH correlated positively with FT-L and FT-MT (r = 0.7371, *p* < 0.001; r = 0.6245, *p* = 0.001 respectively). DMT-1 correlated positively with FT-MT (r = 0.3936; *p* = 0.028), and TRF-1 with FT-H (r = 0.4022, *p* = 0.025). FPN proved to be positively correlated with TRF-1 and L-CH (r = 0.4643, *p* = 0.009; r = 0.5302, *p* = 0.005; respectively). Among iron storage proteins, a positive correlation was found for ferritins: FT-L and FT-MT (r = 0.6626, *p* < 0.001). Also, FT-H correlated positively with FPN (r = 0.6595, *p* < 0.001).

#### 3.3.2. Relations with Regulatory Factors

IREB-2 proved the largest number of correlations with proteins responsible for iron turnover. It presented significant positive correlation with proteins transporting iron into the cell L-CH (r = 0.5198, *p* = 0.007), TFR-1 (r = 0.6008, *<*0.001); iron storage FT-H (r = 0.4660, *p* = 0.008), FT-M (r = 0.4145, *p* = 0.020); and transportation outside the cell FPN (r = 0.6483, *p* < 0.001). ACO-1 revealed positive correlation with TFR-1 (r = 0.3809, *p* = 0.034), FTH (r = 0.4951, *p* = 0.005), and HJV (r = 0.4390, *p* = 0.013). Hepc showed positive correlation with FT-H and HIF-1 (r = 0.4007, *p* = 0.025, r = 0.4282, *p* = 0.016 respectively), HIF-1 with FT-MT (r = 0.3937, *p* = 0.028) and HJV with L-CH (r = 0.4088, *p* = 0.038).

#### 3.3.3. Relations with Oxidative Stress Marker

The oxidative stress marker HNE-4 presented statistically significant negative correlation with TRF-1 (r = −0.4596, *p* = 0.018), FT-MT (r = −0.4148, *p* = 0.035), FPN (r = −0.5976, *p* = 0.001) and IREB-2 (r = −0.5620, *p* = 0.003) (Figure 2, Appendix A).

### 3.4. Correlations between Iron-Handling Proteins, Regulatory Factors and an Oxidative Stress Marker in The Failing Human Myocardium

In FH, we found several correlations among assessed proteins, regulatory factors and oxidative stress.

#### 3.4.1. Relations among Iron Regulatory Proteins

Among the proteins responsible for iron transportation into the cell, a positive correlation was found between DMT-1 and L-CH or TRF-1 (r = 0.6545, *p* < 0.001; r = 0.4456, *p* < 0.001, respectively). Additionally, TFR-1 positively correlated with L-CH (r = 0.3954, *p* = 0.028), DMT-1 positively correlated with all proteins responsible for iron storage FT-H, FT-L, FT-MT (r = 0.2854, *p* = 0.030; r = 0.2842, *p* = 0.031; r = 0.3407, *p* = 0.009, respectively). Also, L-CH positively correlated with FT-H and FT-L (r = 0.4309, *p* = 0.016; r = 0.4909, *p* = 0.005, respectively) and TFR-1 positively correlated with FT-H (r = 0.3627, *p* = 0.005). Moreover, a positive correlation was found between TRF-2 and FT-H or FT-MT (r = 0.4116, *p* = 0.001; r = 0.2916, *p* = 0.026, respectively). Both ferritins FT-H and FT-L also correlated positively (r = 0.5119, *p* < 0.001). FPN positively correlated with all proteins responsible for iron transportation: DMT-1, L-CH, TFR-1 (r = 0.3426, *p* = 0.008; r = 0.5214, *p* = 0.003; r = 0.3804, *p* = 0.003, respectively).

#### 3.4.2. Relations with Regulatory Factors

IREB-2 correlated with DMT-1 (r = 0.4261, *p* = 0.001), TRF-1 (r = 0.2644, *p* = 0.045), FT-H (r = 0.2815, *p* = 0.032), FT-M (r = 0.5455, *p* < 0.001). ACO-1 correlated positively with DMT-1 (r = 0.3624, *p* = 0.005), L-CH (r = 0.3894, *p* = 0.030) and FT-M (r = 0.3277, *p* = 0.012). HIF-1 was positively correlated with DMT-1 (r = 0.5123, *p* < 0.001), L-CH (r = 0.5682, *p* = 0.001), FT-L (r = 0.2597, *p* = 0.049) and FT-M (r = 0.3648, *p* = 0.005), while HJV with DMT-1 (r = 0.3104, *p* = 0.018), L-CH (r = 0.4971, *p* = 0.004) and TRF-1 (r = 0.5472, *p* < 0.001). Hepc correlated with L-CH (r = −0.4626, *p* = 0.009) and FPN (r = −0.2833, *p* = 0.031). We also found significant correlations among regulatory factors: IREB-2 revealed positive correlation with ACO-1 (r = 0.5804, *p* < 0.001), whileACO-1 revealed positive correlation with HIF-1 (r = 0.2639 *p* = 0.045). Hepc and HJV did not correlate with any regulatory factor.

#### 3.4.3. Relations with Oxidative Stress Marker

HNE-4 presented significant negative correlations with: proteins responsible for iron transportation into the cell: DMT-1 (r = −0.3773, *p* = 0.036), L-CH (r = −0.5707, *p* = 0.001); storage protein FT-H (r = −0.3790, *p* = 0.036) and removal protein FPN (r = −0.4219, *p* = 0.018). HNE-4 correlated negatively with ACO-1 (r = −0.4478, *p* = 0.012). (Figure 3, Appendix A).

### 3.5. Correlations between Iron-Handling Proteins and Clinical Data

In FH group, we assessed the relation between iron-related proteins and commonly used systemic iron markers. Only, we found positive correlations among serum ferritin and FT-L/FT-H chains (r = 0.3725, *p* = 0.004; r = 0.4329, *p* < 0.000; respectively), also sTfR with FT-MT (r = 0.3425, *p* = 0.0149). We did not confirm correlations among the rest of the iron-related proteins with other serum iron markers. Due to a lack of iron-related serum markers, we did not assess such relations in NFH.

With regard to heart failure parameters, we only found positive correlations between the FT-MT and CO (r = 0.2742, *p* = 0.0408) and sTfR and CO (r = 0.2646, *p* = 0.0488) (Appendix A).

## 4. Discussion

Iron is a crucial micronutrient for the proper functioning of the heart. The element is involved in oxygen delivery and erythropoiesis, and its deficiency causes anemia and disorder of the oxidative metabolism and cellular immune mechanisms [10,11]. Multiple physiologic (i.e., bleeding, excessive menstruation, malabsorption), environmental (i.e., consumption of foods rich in absorbable iron, intestinal infestations, virus infections) or genetic conditions (i.e., iron-refractory iron deficiency anemia) may cause iron deficiency in humans [12,13]. Iron deficiency may also result in chronic inflammation as an effect of the action of proinflammatory cytokines (IL-6, IL1β, and IL-22), and finally increase hepcidin expression [14]. Iron deficiency in circulation also manifests in the myocardium tissue, and is highly correlated with impaired oxygen consumption and metabolism, especially in heart failure patients [15]. It is estimated that approximately 50% of patients with HF have iron deficiency [16]. Several studies and clinical trials have demonstrated that iron supplementation in HF patients has improved their quality of life and functional capacity, significantly reducing the risk of first hospitalization and hospital readmissions, and improving peak oxygen consumption and functional capacity (measured by a 6-min test walk) [17]. However, some of cardiologists question the beneficial role of iron supplementation in the course of HF, pointing to the lack of significant improvement in the patient’s condition or lack of reduction in the risk of cardiovascular death [4,18]. Moreover, iron overload by supplementation is toxic to the organism. It induces damage in the heart, liver, thyroid, pancreas, and the central nervous system, promoting intensive production of reactive oxygen species (ROS) [19].

Being one of the significant microelements necessary for an organism, iron has a specific system of proteins regulating its concentration in the blood and tissues, and especially in the heart. Therefore, any disturbance in the proteins’ concertation or function may cause iron deficiency. Here we hypothesize that iron deficiency in the heart might be associated with changes in the concentration of proteins involved in iron metabolism in heart failure. Therefore, we compared the concentration of select iron proteins involved in transport (DMT-1, L-CH, TfR-1;-2, FPN); storage (FT-H FT-L, FT-MT); regulation (Hepc, ACO-1, IREB-2, HIF-1, HJV); and a marker of oxidative stress (4-HNE) in the hearts of healthy and HF patients.

DMT-1 is a major transporter of iron II ions (Fe^2+^) in several cells, including myocytes, whereas ferroportin is a sole protein responsible for iron release from cells to blood, where iron is sequestered by transferrin. Cellular intake of iron is also associated with transferrin -1 receptors that bind transferrin, forming a transferrin/transferrin complex entering the target cell via endocytosis [20]. The L-type calcium channel plays an essential role in iron transport in the heart as a high-capacity pathway of ferrous iron (Fe2+) uptake into cardiomyocytes [21,22]. Moreover, there is evidence that L-type and T-type calcium channels may be the main routes of iron uptake to cardiomyocytes under iron overload conditions. Therefore, inhibition of these channels is one of the treatment strategies in patients with iron overload cardiomyopathy [23,24]. Unfortunately, information regarding iron transport within human heart cells is very scarce and yields conflicting data. Thus, reduced expression of TfR1 was reported on an mRNA level, whereas on the protein level, no significant changes were observed, however, on a small population [25,26,27].

Hence, to the best of our knowledge, we are the first to comprehensively report changes in the expression of iron proteins responsible for iron homeostasis in human FH and NFH hearts. We found a significantly lower expression of DMT-1, TfR-1 and L-CH proteins in the heart failure group than in the control group. Downregulation of these proteins was mainly described in iron overloaded hearts. Zhang et al. showed depletion of iron overload in cardiac dysfunction in rats treated with *Salvia miltiorrhiza* injection. The proposed mechanism was associated with a decrease in expression of iron uptake proteins (DMT-1 and TfR-1) and an increase in expression of ferroportin in the heart [28]. Also, Rhee et al. observed that inhibition of DMT-1 function by Ebselen (selective inhibitor of DMT-1) prevented iron-overload associated cardiotoxicity in human iPSC cells [29]. Moreover, genetically modified TfR-1-null mice developed early, lethal cardiomyopathy with heart failure [30]. Our results showing downregulation of iron transporter proteins in HF hearts are in line with previously published work. Maeder et al., in a study of 9 healthy controls and 25 patients with advanced HF, showed a significant reduction in myocardial expression of TfR-1 mRNA [25]. Moreover, a recent study by Tajes and colleagues performed in C57BL/6 mouse with induced HF by isoproterenol osmotic pumps revealed a significant reduction of DMT-1, TfR-1, FT-MT, FPN, Hepc and IREB-1 -2 in both mRNA and protein levels [31]. These downregulations of iron metabolism proteins in iron overload and HF may be associated with an oxidative state of myocardial cells, but more studies are necessary to prove this hypothesis. On the other hand, in HF group the expression of FPN and TfR-2 were unchanged. FPN is responsible for iron release from cells. Deletion of the FPN gene results in fatal left ventricular dysfunction. Its proper action is also necessary for removal of iron from the cardiomyocytes. In ferroportin-knockouts mice, downregulation of TfR1, an IRP-driven response, did not prevent toxicity after iron overload [32]. On the other hand, impaired functioning of TfR-2 causes overproduction of peptide hormone hepcidin and a decrease of iron uptake from the diet [33]. Taken together, lower expression of proteins that deliver iron to the cells and the lack of differences in FPN and TfR-2 proteins expression may be one of the mechanisms of iron deficiency in heart failure.

Iron transport proteins are not the only ones affected by HF. The other major group are proteins responsible for iron storage—ferritins—consisting of two chains: heavy (FT-H) and light (FT-L) [34]. Iron ions may also be stored in the mitochondria by FT-MT [35]. The concentration of ferritins in the tissue is a marker of intracellular iron ions bioavailability. Measurement of ferritin in the blood is the most sensitive and specific indicator of iron deficiency [36]. Low ferritin level in the serum (especially FT-H) is significantly associated with increased risk of heart failure in humans [37]. Also, decreased or lack of FT-H in cardiomyocytes (observed in mice with FT-H gene knockout) enhances oxidative stress, significantly increasing the risk of cardiomyopathy and heart failure [38]. By contrast, iron derived from ferritin induces cardiomyocyte death and heart failure in a process called ferroptosis. Interestingly, ferritins play a protective role against ferroptosis. Mice with FT-H-deficient cardiomyocytes showed increased cardiomyopathy associated with a reduced level of Slc7a11 and glutathione, and increased lipid peroxidation. Our previous studies revealed a reduction in expression ferritin as a whole [26,27]. Here, we analyzed the concentration of different ferritin chains in healthy and failing hearts. The expression of both ferritin chains was significantly lower in the hearts of HF patients, which in turn may enhance myocyte damage in patients supplemented with iron.

An important player in iron homeostasis regulation in the organism is hepcidin. Hepcidin is a 25-aa peptide synthesized mainly by the liver in response to iron binding by transferrin. Transferrin bound with iron ions is called holo-transferrin. When holo-transferrin concentration is increased, it is bound by TfR-1 followed by the release of the human hemochromatosis protein (HFE). Free HFE promotes development of a complex composed of TFR2, HFE, hemojuvelin (HJV), bone morphogenic protein (BMP) and bone morphogenic protein receptor (BMPR), and then through the SMAD pathway triggers the expression of hepcidin [20]. Hepcidin reduces the absorption of iron ions from diet and induces FPN degradation in effector cells, causing a reduction of iron content in the blood. Hecpidin concentration may also be enhanced by inflammation and suppressed by erythropoietic factors [39]. In heart cells, hepcidin works similarly and affects FPN degradation, which should prevent iron sequestration from the heart and the development of ID [40]. Loss of hepcidin in Hepc-knockout mice resulted in fatal contractile and metabolic dysfunction due to cardiomyocyte iron deficiency [41]. Here, we did not observe significant differences in Hepc and HJV protein concentration between HF and non-HF hearts. On the contrary, Janowska et al. observed reduced hepcidin and FPN levels in the circulating blood of high-severity HF patients [42]. These results may indicate that iron metabolism controlled by hepcidin in the heart differs from the hepcidin-dependent iron controlling mechanism in the whole body. We also observed a significant reduction of ACO-1 (also known as IREb-1) and IREB-2 proteins in HF patients. IREBs proteins are responsible for intracellular iron turnover. Low cytosolic iron levels activated ACO-1 and IREB-2, which blocked the translation of FT-L and FT-H and increased the translation of TfR-1 and DMT-1, leading to increased iron import and decreased iron escape from the cells [43]. Mice with a knockout of IREB-1 and -2 genes develop cardiac ID associated with impaired mitochondrial respiration and adaptation to acute and chronic work [44]. Decreased IREB-1 and -2 in patients with HF may be a symptom of a significant disturbance of iron metabolism in cardiomyocytes.

HIF-1 is a heterodimeric transcription factor consisting of two subunits—HIF-1α and HIF-1β. It regulates more than a hundred target genes involved in oxygen homeostasis. Under hypoxic conditions, the activation of the HIF system has a cardioprotective effect and allows the maintenance of the function of the heart muscle [45]. HIF-1 activity in cardiomyocytes is required to maintain oxygen homeostasis during compensatory myocardial hypertrophy. However, it may not be the case in prolonged hypoxia, as shown in models of pressure overload [46]. Prolonged pressure overload leads to heart failure by an accumulation of p53 and impairs cardiac angiogenesis and systolic function, thus leading to chronic heart failure [47]. The HIF system in human end-stage HF myocardium was investigated by Zolt et al., who observed significantly decreased HIF-1 α mRNA in left ventricular myocardium from explanted failing hearts compared to non-failing donor hearts [46]. We also tested HIF-1 in the end stage of HF and found a significant reduction in HIF-1 protein level compared to NHF. However, it is worth emphasizing that HIF-1 α is only one of the elements of the HIF system, and its decreased expression does not indicate the inhibition of the entire pathway.

4-HNE, an α,β-unsaturated hydroxyalkenal, is formed by lipid peroxidation from unsaturated fatty acids, particularly by cells under oxidative stress. The heart is the organ with the highest oxygen uptake and consumption in the body due to its constant activity. The high oxidative capacity of the heart muscle is a source of ROS formation and secondary 4-HNE accumulation. Increased 4-HNE production is an important factor in the pathogenesis of several human diseases, including cancer, neurodegenerative and cardiovascular disease [48,49]. Experimental administration of 4-HNE in animal models led to contractile failure and proarrhythmic effects in hearts [48]. Previous studies have demonstrated that oxidative stress is elevated in the myocardia of patients with heart failure [50]. Nakamura et al. observed that expression of the 4-HNE-modified protein in a myocardial biopsy from patients with dilated cardiomyopathy was significantly increased compared to controls. After treatment with carvedilol, 4-HNE levels decreased by 40% and functional amelioration of heart failure symptoms was observed [50]. Here, we also detected significantly increased 4-HNE levels in the myocardium of HF tissues. 4-HNE is very reactive and generally toxic, so its accumulation leads to the damage of many macromolecules. Our study observed that high 4-HNE levels negatively correlated with the expression of analyzed proteins in the heart muscle.

Furthermore, we analyzed the relationship between iron metabolism proteins in non-HF and HF hearts. This part of the study aimed to evaluate if any relations in protein concentration may pinpoint potential novel molecular targets for the treatment of HF. In non-HF hearts, the highest number of significant correlations (*p* < 0.05) was found for FT-MT, IREB-2, and L-CH, FT-H and FPN. The highest number of moderate/strong correlation r ≥ 0.5, *p* < 0.05 (according to Akoglu 2018 [51]) we found for L-CH, FPN, IREB-2 (4). From proteins regulating iron transport, L-CH and TfR-1 have the highest number and r values of correlations, and these proteins correlated with each other. Moreover, L-CH and TfR-1 proteins were significantly correlated with iron storage (FT-H, FT-MT), iron removal (FPN) and iron regulatory proteins (ACO-1, IREB-2). The correlations were positive, which meant that when one of the proteins’ expressions was low, others were also low. So, the correlations may indicate homeostasis in healthy hearts between iron transportation, storage, removing and regulatory proteins. There was only one protein—TfR-2—without any significant correlation, indicating that the control of hepcidin-dependent iron metabolism is independent. Interestingly, 4-HNE was the only protein in non-HF hearts, negatively correlated with other proteins from iron metabolism (TfR-1, FT-MT, FPN, and IREB-2).

In HF hearts, the highest number of significant correlations (*p* < 0.05) we found for: DMT-1 (11); L-CH (10); and TfR-1, FT-H, ACO-1. The highest number of moderate/strong correlations r ≥ 0.5, *p* < 0.05 we found for L-CH (4). The main correlations between iron transport, storage and regulatory proteins were positive, except hepcidin, which was negatively correlated with L-CH and FPN. Similar to non-HF, all the significant correlations between 4-HNE and other iron metabolism proteins were negative.

## 5. Conclusions

Here, we confirmed the significant increase in oxidative stress in failing human hearts. Moreover, the oxidative stress marker negatively correlated with iron-gathering proteins in non-failing and failing human myocardium. Therefore, the decreased protein expression shown in the study, particularly ferritins, which are responsible for iron buffering, should always be considered. Although iron administration in ID patients has proved to be clinically beneficial, presented results indicate that a careful evaluation of iron metabolism parameters should always be performed prior to iron supplementation.

## 6. Limitation of the Study

As a control (NFH), we used hearts from brain-injured donors. However, NFHs are not necessarily representative of the average human population. These hearts did not receive the same treatment as the hearts from the HF group. Moreover, they received a low dose of dopamine. As the investigated patients presented only with advanced HF, the results from our study should be interpreted cautiously. We cannot rule out that iron load and homeostasis might be somehow different in mild or moderate systolic dysfunction.

## Figures and Tables

**Figure 1 jcm-11-00837-f001:**
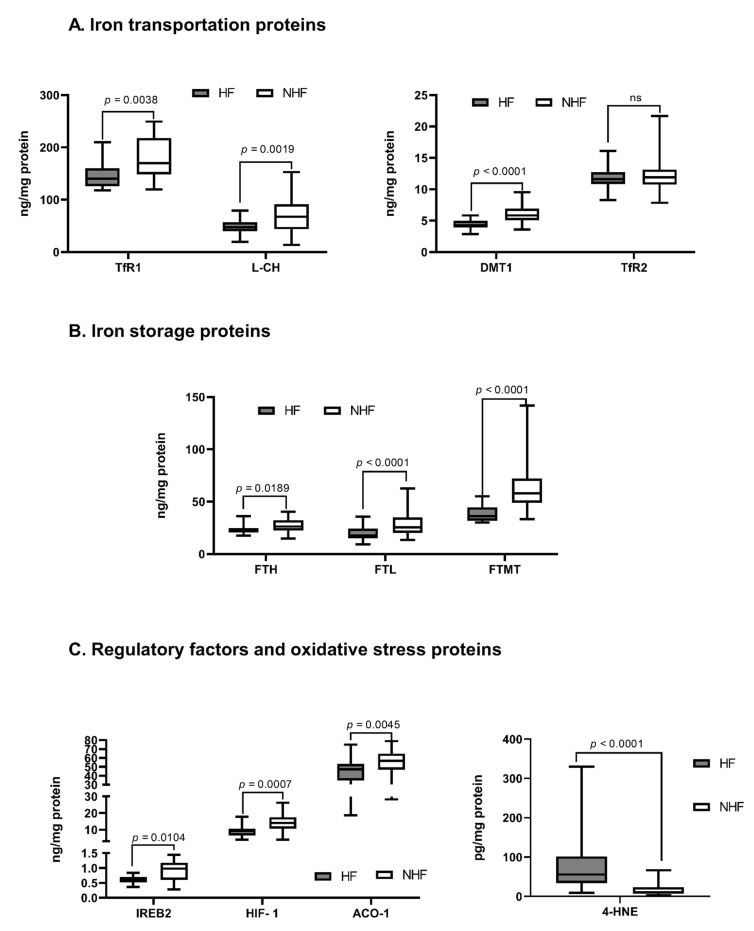
The concentration of iron metabolism proteins in the non-failing human myocardium. (**A**)—Iron transportation proteins, (**B**)—Iron storage proteins, (**C**)—Regulatory factors and oxidative stress proteins. Only significant differences are shown; all results are presented in the supplement (Appendix A); p—level of significance. Aberrations: HF (failing hearts), NHF (non-failing hearts), DMT-1 (divalent metal transporter -1), L-CH (L-type calcium channel Subunit CACNa1D), TfR-1 or -2 (transferrin receptor -1 or -2), FT-H (ferritin heavy chain), FT-L (ferritin light chain), FT-MT (ferritin- mitochondrial), FPN (ferroportin), Hepc (hepcidin), ACO-1 (aconitase- 1), IREB-2 (iron response protein-2), HIF-1 (hypoxia-induced factor), HJV (hemojuvelin), 4-HNE (4-hydroxynonenal).

**Figure 2 jcm-11-00837-f002:**
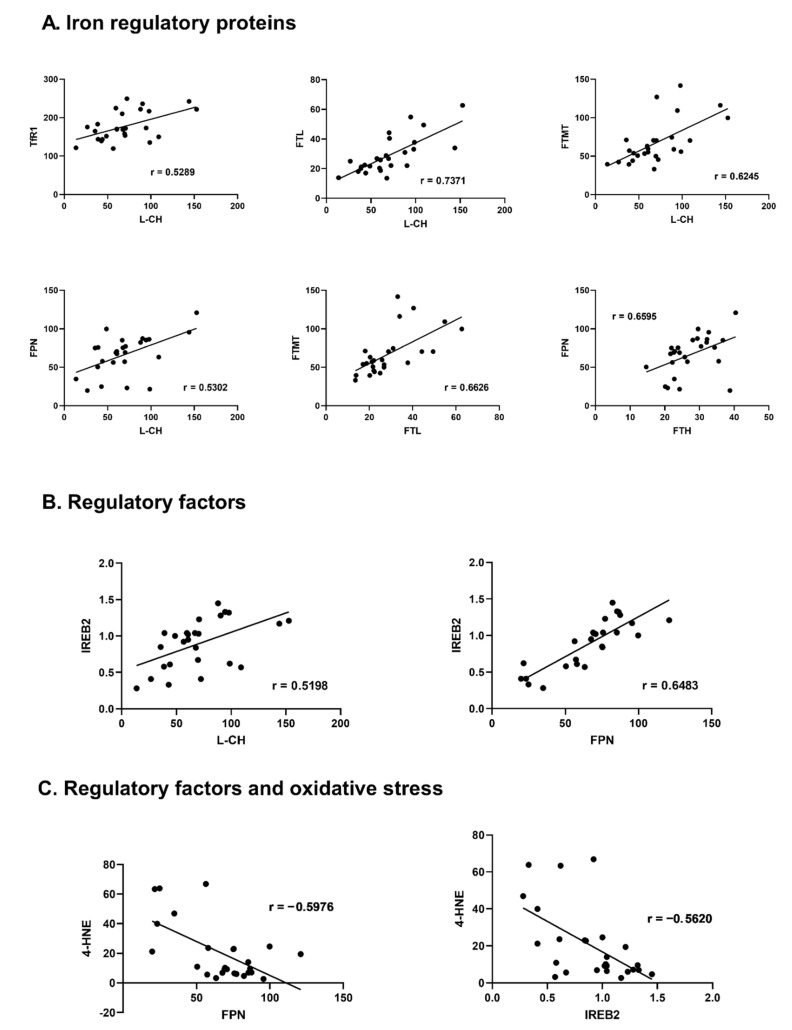
Correlations between iron-handling proteins, regulatory factors and oxidative stress markers in the non-failing human myocardium. (**A**)—Iron regulatory proteins, (**B**)—regulatory factors, (**C**)—Regulatory factors and oxidative stress. Only significant, highly-correlated proteins (r > 0.5, *p* < 0.05) are shown, all correlations are presented in the supplement (Appendix A); r—Pearson correlation coefficient. Aberrations: L-CH (L-type calcium channel Subunit CACNa1D), TfR-1 (transferrin receptor -1), FT-H (ferritin heavy chain), FT-L (ferritin light chain), FT-MT (ferritin- mitochondrial), FPN (ferroportin), IREB-2 (iron response protein-2), HIF-1 (hypoxia-induced factor), 4-HNE (4-hydroxynonenal).

**Figure 3 jcm-11-00837-f003:**
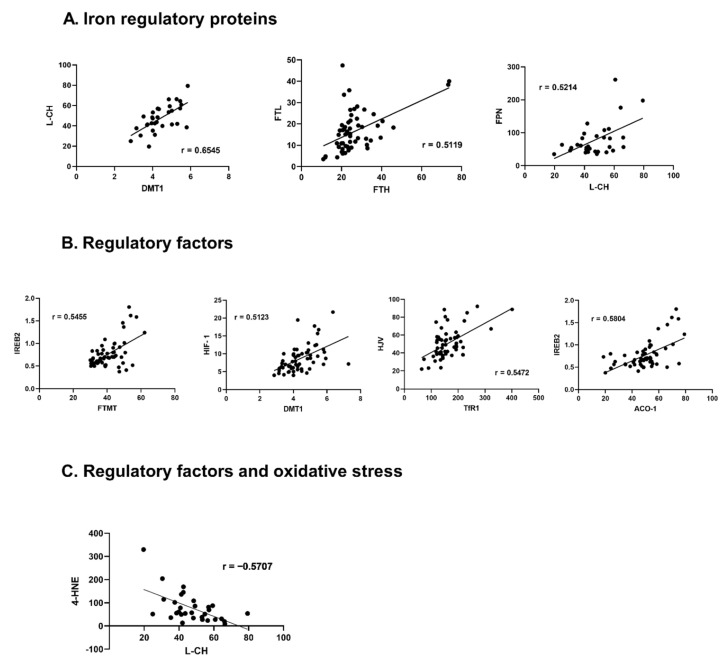
Correlations between iron handling proteins, their regulatory factors, and an oxidative stress marker in the failing human myocardium. (**A**)—Iron regulatory proteins, (**B**)—regulatory factors, (**C**)—Regulatory factors and oxidative stress. Only significant, highly-correlated proteins (r > 0.5, *p* < 0.05) are shown, all correlations are presented in the supplement (Appendix A); r—Pearson correlation coefficient. Aberrations: HF (failing hearts), NHF (non-failing hearts), DMT-1 (divalent metal transporter -1), L-CH (L-type calcium channel Subunit CACNa1D), TfR-1 (transferrin receptor -1), FT-H (ferritin heavy chain), FT-L (ferritin light chain), FT-MT (ferritin- mitochondrial), FPN (ferroportin), ACO-1 (aconitase- 1), IREB-2 (iron response protein-2), HIF-1 (hypoxia-induced factor), HJV (hemojuvelin), 4-HNE (4-hydroxynonenal).

**Table 1 jcm-11-00837-t001:** (**A**,**B**). Clinical characteristics of the study group. A—demographic information, laboratory tests, echocardiography parameters, hemodynamic parameters, iron-related markers. B—etiology. Me—median. Q—quartile; N—number of patients, *p*—level of significance; **bold**—significant differences.

**(A)**								
	**Unit**	**Failing Heart Group (*n* = 58)**		**Women (*n* = 13)**		**Men (*n* = 45)**		***p*-Value**
		**Me**	**Q**	**Me**	**Q**	**Me**	**Q**	
**Demographic Information**								
Age	Yrs	53.5	10.5	58	10	53	10.5	0.4728
Height	Cm	175.5	6.8	164	3.5	178	5	0
Body weight	Kg	80	10.5	61	11.5	82.5	9.5	0.0005
BMI- Body-mass index	kg/m^2^	25.6	3.3	22.8	4.1	25.9	2.6	0.1454
Laboratory tests								
NTproBNP	pg/mL	3955.5	2250.5	4436	1291.5	3833	2665	0.4014
CRP	mg/dL	0.4	0.3	0.3	0.2	0.5	0.3	0.5101
Creatinine	mg/dL	1.2	0.2	1	0.1	1.2	0.2	0.0376
Glomerular filtration rate	ml/min/1.73 m^2^	60	4.5	60	5	60	4.3	0.557
Echocardiography parameters								
LVED	Mm	67	8	58.5	10.5	70	8.5	0.0069
L*VS*D	Mm	59	13	43	10.5	65	12	0.0034
LVEF	%	20	7.5	20	10	20	5.5	0.1812
RV	Mm	42	6.5	42	11	42	5.5	0.8857
IM	+/++++	1.5	0.5	1.5	0.5	1.5	0.5	0.8521
IT	+/++++	1.8	0.8	2.5	0.5	1.5	0.5	0.0183
R*VS*P	mmHg	47	12.5	45	12.5	47	11	0.6869
TAPSE	Mm	15	2	13	2	16	2	0.0134
Hemodynamic parameters								
PAPs	mmHg	42	12.5	36	8.5	44	12	0.1052
PAPm	mmHg	29	7.5	26.5	5.5	29	7	0.2711
PWPm	mmHg	21	5.5	19.5	6	21	7.5	0.2624
TCG	mmHg	8.5	3	9	1.5	8	3	0.4742
CO	l/min	3.5	0.8	2.6	0.4	3.7	0.7	0.0058
PVR	jWood	2.2	0.9	2.7	1.4	2.2	0.7	0.2157
SVR	jWood	13.3	2.8	22.4	10.9	12.6	2.2	0.0969
Iron-related markers								
HT	g/dL	14.1	1.4	14.3	1.5	14	1.2	0.5261
MCV	Fl	91	3.6	90	4.8	91	3.5	0.7372
RET %	%	1.6	0.5	1.7	0.2	1.6	0.5	0.7642
RET-Hb	Pg	32	2	26	3.1	32	1.6	0.096
Ferritin	ng/mL	212.1	89.1	230	122.5	202.6	82.5	0.8663
TSAT	%	21	8.6	18	7.1	24.2	7.5	0.1138
sTfR	mg/L	3.2	1.1	3.3	1.6	3.2	1.2	0.4381
sTfR1/logFR	-	0.61	0.27	0.63	0.36	0.61	0.21	0.2327
Erythropoietin	mIU/mL	18.3	11.4	28.6	16.1	16.1	10.1	0.0386
Aberrations: NTproBNP (N-terminal pro-brain natriuretic peptide), CRP (C-reactive protein), LVED (left ventricle diameters diastolic), LVSD (left ventricle diameters systolic), LVEF(left ventricle ejection fraction), RV (right ventricle diastolic size), IM (mitral insufficiency), IT(tricuspid insufficiency), RVSP (right ventricular systolic pressure), TAPSE (tricuspid annular plane systolic excursion), PAPs (pulmonary artery pressure systolic), PAPm (pulmonary artery pressure mean), PWPm (pulmonary capillary wedge pressure mean), TCG (transpulmonary pressure gradient), CO(cardiac output), PVR (pulmonary vascular resistance), SVR (systemic vascular resistance),HT (hematocrit), MCV (mean corpuscular volume), RET % (reticulocyte %),RET-Hb (reticulocyte hemoglobin), TSAT (transferrin saturation), sTfR (soluble transferrin receptor) sTfR1/logFR (soluble transferrin receptor/logarithm of ferritin)								
**(B)**								
	**Heart failure**			**Women**		**Men**		***p*-Value**
**Etiology**	**N**	**%**		**N**	**%**	**N**	**%**	
Ischemic	18.0	31.0		2.0	15.4	16.0	35.6	
Cardiomyopathy	36.0	62.1		9.0	69.2	27.0	60	0.2008
Other	4.0	6.9		2	15.4	2	44.4	
NYHA: 3	31	53.4		6	46.2	25	55.5	0.5494
NYHA: 4	27	46.6		7	53.84	20	44.4	
Sinus rhythm	40	69		7	53.84	33	73.3	0.1810
No Sinus rhythm	18	31		6	46.2	12	26.7	
Aberrations: NYH (New York Heart Association)								

## Data Availability

The data presented in this study are available on request from the corresponding author. The data are not publicly available due to founding agreement limitations.

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
