# Peer review of "Expression of Iron Metabolism Proteins in Patients with Chronic Heart Failure"

_jcm, 2022, doi:10.3390/jcm11030837_

Round 1
Reviewer 1 Report
The authors found that iron-related proteins are less in human failing heart. The authors investigated whether there was a link between the amount of iron-related protein in the heart and the amount of 4-HNE.
I have some questions.
- Is the title correct?
- Figure 1. C. Regulatory factors and “oxidate” stress proteins. Oxidative?
- There are previous studies comparing failing and non-failing heart tissue using comprehensive analyses like RNA-seq or proteomics. Are there any data in these studies that support the authors’ data (downregulation of iron-related proteins in failing heart)?
- Please show the association between iron-related markers in the blood and iron-related proteins in heart tissue.
- How many people were diagnosed with iron deficiency and those who were not in each group of FH and NFH?
- Is there any difference in the amount of iron-related proteins in heart tissue between patients with and without iron deficiency ?
- What kind of cells mainly express each iron-related protein? Cardiomyocytes or non-cardiomyocytes? The proportion of these cells may be different between failing and non-failing heart.
- Authors described “Reducing iron-gathering proteins and elevated oxidative stress in failing hearts may lead to increased myocardial iron toxicity, which should be considered before iron supplementation.” This is a leap in theory.
Author Response
Response to the Reviewer
We want to thank you for all your valuable comments that allowed us to improve our manuscript. All changes of earlier text are marked in tracking changes and are included in the current manuscript text according to all reviewer suggestions. Below, we include also all reviewers’ comments with our responses.
First round
Reviewer 1:
- Is the title correct?
Response: we improved the title: Expression of iron metabolism proteins in the patients with chronic heart failure
- Figure 1. C. Regulatory factors and “oxidate” stress proteins. Oxidative
Response: We changed the description on the Fig. 1
- There are previous studies comparing failing and non-failing heart tissue using comprehensive analyses like RNA-seq or proteomics. Are there any data in these studies that support the authors’ data (downregulation of iron-related proteins in failing heart)?
Response: We fully agree with the reviewer – data from RNA-seq in HF is available in the literature. However, the data obtained from these studies are often not similar to each other, due to differences in sample number (from 4 to 64 patients), inclusion criteria, the severity of the heart failure, etc. We have evaluated the data from more than 10 articles, and we present here the summary of the results (table below).
|
Detected protein |
gene |
PMID: 30419824 |
PMID: 28497843 |
PMID: 25528681 |
|
number of samples |
64 |
4 |
6 |
|
|
divalent metal transporter - (DMT-1) |
SLC11A2 |
n.f. |
n.f. |
n.f. |
|
L-type calcium channel Subunit CACNa1D (L-CH) |
CACNA1D |
n.f. |
n.f. |
n.f. |
|
transferrin receptor -1(TfR) |
TFRC |
n.f. |
n.f. |
decreased |
|
transferrin receptor -2 (TfR-2) |
TFR2 |
n.f. |
n.f. |
n.f. |
|
ferritin heavy (FT-H), |
FTH1 |
decreased |
n.f. |
n.f. |
|
ferritin – light (FT-L), |
FTL |
n.f. |
n.f. |
n.f. |
|
ferritin- mitochondrial (FT-MT) |
FTMT |
n.f. |
n.f. |
n.f. |
|
ferroportin (FPN) |
SLC40A1 |
n.f. |
increased |
n.f. |
|
hepcidin (Hepc) |
HAMP |
n.f. |
n.f. |
n.f. |
|
aconitase- 1 (ACO-1) |
ACO1 |
n.f. |
n.f. |
n.f. |
|
iron response protein-2 (IREB-2) |
IREB2 |
n.f. |
n.f. |
n.f. |
|
hypoxia-induced factor (HIF-1) |
HIF1 |
n.f. |
n.f. |
n.f. |
|
hemojuvelin (HJV) |
HFE2 |
increased |
n.f. |
n.f. |
Based on the comments from Reviewer 4, we added some information from protein studies: Our results showing downregulation of iron transporter proteins in HF hearts are in line with previously published work. Maeder et al. in the study of 9 healthy controls and 25 patients with advanced HF showed a significant reduction of myocardial expression of TfR-1 mRNA [25]. Moreover, a recent study by Tajes and colleagues performed in C57BL/6 mouse with induced HF by isoproterenol osmotic pumps revealed a significant reduction of DMT-1, TfR-1, FT-MT, FPN, Hepc, and IREB-1 -2, in both mRNA and protein levels [31]. These downregulations of iron metabolism proteins in iron overload and HF may be associated with an oxidative state of myocardial cells however, more studies are necessary to prove this hypothesis.
- Please show the association between iron-related markers in the blood and iron-related proteins in heart tissue.?
Response: We fully agree with the presented point of view, which additionally enriching the value of our work, so we have carried out the correlation and presented the result in the paragraph 3.5, and the most important selected variables in Table S8. Due to lack of iron-related serum markers we did not assessed such relations in NFH – it was pointed out in paragraph 3.5.
- How many people were diagnosed with iron deficiency and those who were not in each group of FH and NFH?
Response: Within FH subjects we diagnosed 26 subjects Non Iron Deficient and 32 Iron deficient. Due to lack of iron-related serum markers we did not assessed such relations in NFH – it was pointed out in paragraph 3.5.
- Is there any difference in the amount of iron-related proteins in heart tissue between patients with and without iron deficiency
Response: All data with regard to myocardial iron related proteins in FH (26 subjects Non Iron Deficient and 32 Iron deficient) were presented in paragraph 3.2.2 and also table S8.
- What kind of cells mainly express each iron-related protein? Cardiomyocytes or non-cardiomyocytes? The proportion of these cells may be different between failing and non-failing heart.
Response: In HF hearts there are two major populations of the cells: cardiomyocytes and cardiofibroblast. Both of them may express iron metabolism proteins. There is, however little information that compares the level of iron proteins in these cells. Recent work by Kobak et al. (https://doi.org/10.3390/cells10040818) showed similar levels of this protein in cardiomyocytes and cardiofibroblast treated with serum from healthy patients. Interestingly, treatment with serum from myocarditis causes a similar increase in protein levels in cardiomyocytes and cardiofibroblast, which may suggest that both tapes of cells are necessary for iron storage in the heart. Nevertheless, due to the nature of techniques used, our analysis does not provide cell type specific resolution of obtained data therefore, our team is currently validating new methodology that will allow for the separation of these cells into individual populations (cardiac fibroblasts and cardiomyocytes) for further studies.
- Authors described “Reducing iron-gathering proteins and elevated oxidative stress in failing hearts may lead to increased myocardial iron toxicity, which should be considered before iron supplementation.” This is a leap in theory.
Response: We have rewritten the sentence from the abstract. : Reducing iron-gathering proteins and elevated oxidative stress in failing hearts is very unfavorable for myocardiocytes. It should be taken into consideration before treatment with drugs or supplements which elevated free oxygen radicals in heart.

Reviewer 2 Report
This study was conducted by the Department of Heart Failure and Transplantology,the Cardinal Stefan Wyszynski National Institute of Cardioalogy,Warsaw,Poland in collaboration with several other departments.
The study examines expression of iron metabolism proteins in the human chronic heart failure hearts compared to normal hearts. Iron deficiency in the patients with heart failure is very strong negative predictor of poor outcomes in those patients.All major heart failure guidelines have strong recommendations for iron replacement in those patients.Therefore iron metabolism and proteins involved in this process are of great interest to the clinicians.
It is a nice exploratory study but no hypothesis described.
The background of the control group is very limited and we don't know if those patient had medical conditions or treatments that could have affected iron metabolism.
Overall it's a nice study which should help to better understand iron metabolism in failing hearts.
Author Response
Response to the Reviewer 2
We want to thank you for all your valuable comments that allowed us to improve our manuscript. All changes of earlier text are marked in tracking changes and are included in the current manuscript text according to all reviewer suggestions. Below, we include also all reviewers’ comments with our responses.
First round
Reviewer 2:
- It is a nice exploratory study but no hypothesis is described.
Response: Our work was to find differences between HF and non-HF hearts in the iron-metabolism proteins. We hypothesize that the concentration of proteins involved in the iron metabolism in ID HF hearts may be significantly dysregulated compared to those in normal hearts. We also wanted to check, based on the differences in iron-related protein expression in HF and NHF hearts, if there is a potential danger from iron supplementation in patients with chronic heart failure due to iron overload cytotoxicity.
- The background of the control group is very limited and we don't know if those patient had medical conditions or treatments that could have affected iron metabolism.
Response: We are aware that the control group is not very well described mainly due to the special conditions in which the heart is obtained (unexpected death). However, it has to be noted that hearts used for transplantation are used only from healthy patients without any chronic diseases or hematology diseases, which was confirmed with the family members of the deceased.
We added the sentence to M&M section : Donors from the control group did not have chronic diseases, which was confirmed with family members of the deceased.

Reviewer 3 Report
The design of the study is clear. The study is well planned. The research methods are appropriate for the purpose.
Author Response
Response to the Reviewer 3
We want to thank the reviewer for all your valuable work you put on manuscript.
We are very happy that the study is acceptable for you and you accept it for publication.
Once more thank you very much.

Reviewer 4 Report
The current manuscript describes a study which sought to assess the levels of iron- related proteins in non-failing versus failing human myocardium. The authors made correlations among the different studied proteins and added an oxidative stress marker in the analysis. The authors concluded that there was an increased in oxidative stress in failing human heart and a negative correlation between the oxidative stress marker and the iron gathering proteins in the studied human myocardium.
The study is interesting and the samples used very valuable. The limitations are adequately addressed. In general terms, the manuscript can be considered of interest.
However, several issues were found and several suggestions need to be addressed by the authors.
- In the Abstract the authors state: “Reducing iron-gathering proteins and elevated oxidative stress in failing hearts may lead to increased myocardial iron toxicity, which should be considered before iron supplementation”. However, data showing iron toxicity in the samples used or any analyzes in iron-supplemented patients are not included. Therefore, this statement should be included in the Discussion section rather than in the Abstract or in the Conclusion.
- Because of the importance of iron toxicity in the text, iron levels and toxicity should be determined in the studied samples.
- In the Material and Methods section, specifically in the point 2.4, it is mentioned the amount of NP40 used in the case of rat tissue experiment. This point should be clarified.
- Since functional ferritin protein is composed by both heavy and light chains, to avoid misunderstanding the name of “ferritin heavy” and “ferritin light” should be modified by “ferritin heavy chain” and “ferritin light chain” along the full manuscript.
- In the Results section, specifically in the point 3.2, the oxidative stress marker HNE was analysed, showing a big deviation in the failing myocardium. Have the authors, or others, measured another oxidative stress marker in the same type of samples? Since the relationship between oxidative stress and iron toxicity is an important point, this one should be explained in more detail.
- It should be interesting to explore whether myocardial iron-related proteins are related to systemic iron markers (i.e. ferritin, transferrin, TSAT, sTFR...) in both failing and non-failing hearts.
- Are the myocardial iron-related proteins related to echocardiography and hemodynamic parameters?
- In the Discussion section, line 318, it is stated that the downregulation of the studied proteins were mainly describe in iron overloaded hearts. Nevertheless, other studies have shown a similar downregulation in iron depleted hearts (PMID: 34001233, PMID: 21777743). Therefore, this point need to be further discussed including the current controversy.
- Please, unify the criteria along the full manuscript. For instance, in the line 374 it is stated SLC11A2, whereas in the rest of the manuscript the authors use DMT-1.
- In the Conclusion section, the authors highlight the relevance of their study in the iron supplemented patients. But no data on iron supplemented patients are provided supporting this point. Therefore, this point may be discussed in the Discussion section, but not stated as a conclusion.
- Some grammatical errors are present throughout the manuscript.

Author Response
Response to the Reviewer 4
We want to thank you for all your valuable comments that allowed us to improve our manuscript. All changes of earlier text are marked in tracking changes and are included in the current manuscript text according to all reviewer suggestions. Below, we include also all reviewers’ comments with our responses.
First round
Reviewer 4:
- In the Abstract the authors state: “Reducing iron-gathering proteins and elevated oxidative stress in failing hearts may lead to increased myocardial iron toxicity, which should be considered before iron supplementation”. However, data showing iron toxicity in the samples used or any analyzes in iron-supplemented patients are not included. Therefore, this statement should be included in the Discussion section rather than in the Abstract or in the Conclusion.
Response: We have rewritten the Abstract and Conclusion sections.
- Because of the importance of iron toxicity in the text, iron levels and toxicity should be determined in the studied samples.
Response: We fully agree with the need for further studies evaluating assessment of the iron concentration in the studied group, however, it is practically impossible due to the high cost and the long duration of the assay. We are currently applying for another funding and we plan to analyze it in the next research. Iron cytotoxicity is generally well known (https://www.sciencedirect.com/science/article/pii/S0167488916303275). The effect of iron cytotoxicity is associated with an increase in oxidative stress, which was confirmed in the present paper by changes in oxidative-dependent proteins in samples.
- In the Material and Methods section, specifically in the point 2.4, it is mentioned the amount of NP40 used in the case of rat tissue experiment. This point should be clarified.
Response: It was an editorial error. We deleted rat NP-40.
- Since functional ferritin protein is composed by both heavy and light chains, to avoid misunderstanding the name of “ferritin heavy” and “ferritin light” should be modified by “ferritin heavy chain” and “ferritin light chain” along the full manuscript.
Response: We changed it.
- In the Results section, specifically in the point 3.2, the oxidative stress marker HNE was analysed, showing a big deviation in the failing myocardium. Have the authors, or others, measured another oxidative stress marker in the same type of samples? Since the relationship between oxidative stress and iron toxicity is an important point, this one should be explained in more detail.
Response: We agree with the reviewer – results obtained in the present study revealed an important role of oxidation in heart failure. Unfortunately, due to the number of factors investigated in the current study we were unable to evaluate another oxidative stress marker. Knowing the importance of the iron toxicity and based on the results obtained, in future studies, we will take a closer look at the oxidative stress.
- It should be interesting to explore whether myocardial iron-related proteins are related to systemic iron markers (i.e. ferritin, transferrin, TSAT, sTFR...) in both failing and non-failing hearts.
Response: We fully agree with the presented point of view, which additionally enriching the value of our work, so we have carried out the correlation and presented the result in the paragraph 3.5, and the most important selected variables in Table S8. Due to lack of iron related serum markers we did not assessed such relations in NFH – it was pointed out in paragraph 3.5.
- Are the myocardial iron-related proteins related to echocardiography and hemodynamic parameters?
Response: As above we also fully agree with the presented point of view, and we have carried out the correlation, presented the result in the paragraph 3.5, and the most important selected variables in Table S8.
- In the Discussion section, line 318, it is stated that the downregulation of the studied proteins were mainly describe in iron overloaded hearts. Nevertheless, other studies have shown a similar downregulation in iron depleted hearts (PMID: 34001233, PMID: 21777743). Therefore, this point need to be further discussed including the current controversy.
Response: We have described the topic in more detail:
Interestingly, there are also studies that have shown a similar downregulation of iron transported proteins in human and animal HF hearts, observed in the present work. Maeder et al. in the study of 9 healthy controls and 25 patients with advanced HF showed a significant reduction of myocardial expression of TfR-1 mRNA [25]. Moreover, a recent study by Tajes and colleagues performed in C57BL/6 mouse with induced HF by isoproterenol osmotic pumps revealed also revealed a significant reduction of DMT-1, TfR-1 and also FT-MT, FPN, Hepc, IREB-1 -2, in both mRNA and protein levels [31]. These downregulations of iron metabolism proteins in iron overload and HF may be associated with an oxidative state of myocardial cells, however, to prove that future studies should be performed.
- Please, unify the criteria along the full manuscript. For instance, in the line 374 it is stated SLC11A2, whereas in the rest of the manuscript the authors use DMT-1.
Response: We changed it.
- In the Conclusion section, the authors highlight the relevance of their study in the iron supplemented patients. But no data on iron supplemented patients are provided supporting this point. Therefore, this point may be discussed in the Discussion section, but not stated as a conclusion.
Response: We have rewritten two sentences of the Conclusion section: Therefore, the decreased protein expression shown in the study, particularly ferritin, which is responsible for iron buffering, should always be considered. Although iron administration in ID patients proved to be clinically beneficial, presented results indicate that a careful evaluation of iron metabolism parameters should always be performed prior to iron supplementation”.
- Some grammatical errors are present throughout the manuscript.
Response: We improved some grammatical errors, however before printing we will use a native speaker service.
